# Identification of σ^E^-Dependent Promoter Upstream of *clpB* from the Pathogenic Spirochaete *Leptospira interrogans* by Applying an *E. coli* Two-Plasmid System

**DOI:** 10.3390/ijms20246325

**Published:** 2019-12-15

**Authors:** Sabina Kędzierska-Mieszkowska, Katarzyna Potrykus, Zbigniew Arent, Joanna Krajewska

**Affiliations:** 1Department of General and Medical Biochemistry, University of Gdańsk, Faculty of Biology, 80-308 Gdańsk, Poland; joanna.krajewska@ug.edu.pl; 2Department of Bacterial Molecular Genetics, Faculty of Biology, University of Gdańsk, 80-308 Gdańsk, Poland; katarzyna.potrykus@ug.edu.pl; 3University Centre of Veterinary Medicine, University of Agriculture in Krakow, 30-059 Krakow, Poland; zbigniew.arent@urk.edu.pl

**Keywords:** *clpB*, *Leptospira*, leptospirosis, molecular chaperone, sigma factor, transcription

## Abstract

There is limited information on gene expression in the pathogenic spirochaete *Leptospira*
*interrogans* and genetic mechanisms controlling its virulence. Transcription is the first step in gene expression that is often determined by environmental effects, including infection-induced stresses. Alterations in the environment result in significant changes in the transcription of many genes, allowing effective adaptation of *Leptospira* to mammalian hosts. Thus, promoter and transcriptional start site identification are crucial for determining gene expression regulation and for the understanding of genetic regulatory mechanisms existing in *Leptospira*. Here, we characterized the promoter region of the *L. interrogans clpB* gene (*clpB_Li_*) encoding an AAA^+^ molecular chaperone ClpB essential for the survival of this spirochaete under thermal and oxidative stresses, and also during infection of the host. Primer extension analysis demonstrated that transcription of *clpB* in *L. interrogans* initiates at a cytidine located 41 bp upstream of the ATG initiation codon, and, to a lesser extent, at an adenine located 2 bp downstream of the identified site. Transcription of both transcripts was heat-inducible. Determination of *clpB_Li_* transcription start site, combined with promoter transcriptional activity assays using a modified two-plasmid system in *E. coli*, revealed that *clpB_Li_* transcription is controlled by the ECF σ^E^ factor. Of the ten *L*. *interrogans* ECF σ factors, the factor encoded by *LIC_12757* (*LA0876*) is most likely to be the key regulator of *clpB* gene expression in *Leptospira* cells, especially under thermal stress. Furthermore, *clpB* expression may be mediated by ppGpp in *Leptospira*.

## 1. Introduction

*Leptospira interrogans* is one of the many pathogenic species of the genus *Leptospira* and is composed of several strains that can cause leptospirosis in mammals, including humans. It has been reported that over 1 million human cases of severe leptospirosis occur worldwide each year, with approximately 60,000 deaths from this disease [1]. It has to be noted that leptospirosis also generates huge economic losses in a number of countries due to reproductive disorders in cattle, sheep, pigs, and horses [2,3]. Furthermore, recent serological and microbiological studies have indicated a high rate of leptospiral infections in domestic animals [2,4,5,6]. Despite a high risk of leptospirosis, especially in the tropical and subtropical countries, and its global importance, molecular mechanisms of both the leptospiral virulence and the disease pathogenesis currently remain largely unknown [7,8], mainly due to the historical lack of standard genetic tools for use in work with the pathogenic *Leptospira* species. Recent advances in genetic manipulation of these species have made it possible to identify several leptospiral virulence factors. However, many of them have turned out not to be required for virulence in animal models [7]. Unfortunately, the limitations of modern genetic tools available for pathogenic *Leptospira* spp. still have an enormous impact on the understanding of the molecular and cellular mechanisms involved in the pathogenesis of leptospirosis.

Like other pathogenic bacteria, *L. interrogans* is exposed to environmental stresses during infection of mammalian hosts. Bacteria fight various environmental stressors by altering the expression of genes involved in host adaptation and promoting their survival. Transcriptional regulation, and especially sigma factors controlling the promoter selectivity of bacterial RNA polymerase, play a crucial role in this stress-induced gene expression response. Unfortunately, there is limited information on gene regulation in *Leptospira* spp. Comparative genomics and genome-wide in silico analyses had shown that the *Leptospira* genome contains one basic sigma factor-encoding gene- *rpoD* (σ^70^), and alternative sigma factor genes: *rpoF* (σ^28^), *rpoN* (σ^54^), and several (5–11) genes encoding extracytoplasmic function (ECF) sigma factors, referred to as *rpoE* (σ^E^) in the literature [9,10] (Table 1).

According to these analyses, the housekeeping sigma factor, σ^70^, coordinates transcription of most genes in the leptospiral cells, while alternative sigma factors play key roles in environmental adaptation of bacteria and their virulence. In that light, it is intriguing that the *Leptospira* species differ in the number of ECF σ factors. For example, pathogenic *L. interrogans* encodes ten different ECF σ factors (Table 2), while saprophytic species encode only five [9].

On the other hand, the well-known bacterium *E. coli* possesses only two ECF σ factors [14] (Table 1). It is postulated that the ECF σ factors existing only in the pathogenic *Leptospira* species are important for the *Leptospira* life cycle within the host [9]. Unraveling regulation of virulence genes’ expression is a particularly important challenge because it is necessary for understanding the molecular basis of the disease caused by the pathogen and its underpinning. Molecular chaperone ClpB, a member of the Hsp100/Clp subfamily of the AAA + ATPases (*ATPases associated with a variety of cellular activities*), is among the known leptospiral virulence factors. It has been demonstrated that ClpB deficiency in *L. interrogans* resulted in bacterial growth defects under oxidative and heat stresses, and also in its loss of virulence [15]. Furthermore, in previous studies, we had shown that *L. interrogans* ClpB (ClpB_Li_) is not only synthesized but is also immunogenic during the infection process, further supporting its involvement in *Leptospira* pathogenicity [16]. Our recent studies suggest a possible role of ClpB_Li_, i.e., its aggregate-reactivation activity is necessary for maintaining the energy-generating metabolism of the *Leptospira* cell [17], again strongly supporting ClpB’s importance in leptospiral virulence and implying the importance of *clpB* gene expression regulation. Here, primer extension analysis combined with promoter activity assays using a genetic strategy revealed that *clpB_Li_* transcription is σ^E^ dependent. We postulate that factor encoded by *LIC_12757* (*LA0876*) is a key ECF σ^E^ factor that promotes *clpB* transcription in pathogenic *Leptospira*, especially under stressful conditions. In addition, the ppGpp alarmone may act as a regulator of *clpB* expression in those cells. Our findings provide the first insight into *Leptospira clpB_Li_* transcription regulation. Still, further studies are needed to discover all regulatory elements affecting *clpB* expression in *Leptospira* cells.

## 2. Results and Discussion

### 2.1. Identification of the clpB Transcriptional Start Sites by Primer Extension

In order to determine *clpB* transcriptional start sites and identify functional promoters responsible for its expression in *Leptospira*, primer extension assays were performed. Three synthetic oligonucleotides, p_51–72_, p_290–321_, p_565–591_ (see Table 3) were hybridized to RNA extracted from both, *Leptospira interrogans* and *E. coli* MC4100Δ*clpB* cells carrying the *L. interrogans clpB* gene (*clpB_Li_*) cloned into a low-copy pGB2 plasmid together with a 500-bp DNA fragment containing sequence located upstream of the gene (pClpB_Li_). A BLAST analysis did not show a potential cross-hybridization of the designed primers with regions of the *E. coli* genome, suggesting high specificity of the designed primers. To prepare RNA, the *L. interrogans* strain was grown at 28 °C for optimal in vitro growth and also under thermal stresses at 37 and 42 °C for 4 and 2 h, respectively, whereas *E. coli* cells were grown at 30 or 42 °C for 30 min (heat shock). Following reverse transcription and analysis on a sequencing gel, one major product was observed when the p_51–72_ primer was used in reactions with RNA isolated from *Leptospira* cells grown at all tested temperatures (Figure 1). This product corresponds to a transcriptional start site at a cytidine residue located 41 bp upstream of the first *clpB* ATG codon (−41 relative to ATG). In addition, a lesser band is observed at an adenosine residue located at the −39 position relative to ATG. It is interesting to note that this alternative start point of transcription (A-39) coincides with the primary transcriptional start site (tctAaac) previously predicted by [10] using differential RNA-sequencing (dRNA-seq). It is also worth mentioning that the secondary transcriptional start site (G-526 of the GenBank *clpB* gene sequence, accession number M28364) was found to be close to the primary transcriptional start site (A-525) in primer extension analysis of the *E. coli clpB* transcripts [18]. It is also known that *Mycoplasma pneumoniae* reference strain M129 uses two different transcriptional start sites (G-10 and A-7 relative to the ATG initiation codon) to produce ClpB [19]. Thus, the presence of the primary and secondary transcriptional start site in the *clpB_Li_* promoter region is not surprising. Further, the amount of both *Leptospira clpB* transcripts is increased after exposure of cells to a temperature of 42 °C for 2 h. This result indicates that *clpB_Li_* expression in *Leptospira* is induced by temperature stress, which is consistent with the findings of [15], who demonstrated that *clpB_Li_* transcription in *Leptospira* had increased by 3-folds after 2 h exposure to heat stress. One possible candidate regulating this expression could be RpoH (σ^32^; heat shock sigma factor); however, comparative genomics and genome-wide in silico analyses had shown no presence of a *rpoH* gene in the *Leptospira* genome [9,10]. Thus, *Leptospira* spp. does not possess σ^32^ that is required for heat shock gene expression and heat shock response in *E. coli*. On the other hand, as shown in Figure 1, for RNA extracted from the *E. coli* cells, no extension product was detectable with the p_51-72_ primer, indicating that *E. coli* either does not possess an appropriate factor for *clpB_Li_* expression or this factor is not induced in *E. coli* by temperature stress. Additional analysis has been performed with two other primers. Primer extension assay utilizing the p_290–321_ primer did not detect any transcripts either with *Leptospira* or with *E. coli* RNA (not shown), while the use of the p_565–591_ primer revealed a few weak heat-inducible transcripts for *Leptospira* and one transcript for *E. coli* RNA that was not induced by heat shock stress and did not coincide with any leptospiral transcription start sites (Figure 1). All of these transcripts initiate within the region that is 20–96 nt upstream of a potential internal translation initiation site of *clpB_Li_* with a good Shine–Dalgarno sequence (AGGGAA) and the ATG codon (see Figure 1). It is known that *E. coli* uses two translational start sites to produce two isoforms of ClpB, the full-length ClpB95 and shorter ClpB80, which does not contain the substrate-interacting N-terminal domain [18,20,21]. It has been demonstrated that the two isoforms of ClpB from *E. coli* cooperate in reactivation of aggregated proteins to form a highly efficient chaperone system [21]. It is very likely that the functional cooperation between ClpB isoforms arises from interactions between them because it was found that ClpB95 and ClpB80 associate into hetero-oligomers, which boost the aggregate-reactivation potential of the ClpB chaperone [21]. Thus, it was postulated that *E. coli* produces two isoforms of ClpB to optimize its disagregase activity. The presence of two forms of ClpB, a 96-kDa and an 80-kDa protein, was also demonstrated in *L. interrogans* [15]. Our results point to the occurrence of an additional transcriptional start site located within the *clpB_Li_* coding region in both *E. coli* and *Leptospira*. Still, it is intriguing that even though the internal *clpB_Li_* transcriptional start sites were detected in both instances, they differ depending on the host. It is also interesting to note the similarity between one of the internal transcriptional start sites (see Figure 1, gagtTtt) that we mapped by primer extension analysis and one of the sixteen internal start points (gagTttt) previously predicted by [10]. Thus, it is highly possible that *clpB_Li_* has an additional promoter. On the other hand, it cannot be excluded that the mapped minor transcripts result from the processing of the major transcript produced. Still, further studies are needed to examine the nature of those minor transcripts in *Leptospira*.

### 2.2. Presence of a Promoter Dependent on σ^E^ Upstream of the ClpB_Li_ Gene and its Activity in E. coli Cells

After the mapping of the two *clpB_Li_* transcriptional start sites (both probably originating from the same promoter) in *Leptospira* (see Figure 1), the nucleotide sequence upstream of *clpB_Li_* was carefully investigated. A putative σ^E^ promoter element was successfully identified on the basis of sequence similarity to the well-characterized *E. coli* σ^E^ core promoters. As shown in Figure 2, sequences of the −10 and −35 motifs of the potential *clpB* promoter (*clpB_Li_P1*) exhibit a considerable homology to the known promoters controlled by *E. coli* σ^E^.

To investigate whether the σ^E^ factor may be really involved in the *clpB_Li_* transcriptional regulation, promoter activity assay was carried out under σ^E^-limiting (basal) and σ^E^ excess (overproduction) conditions. It is worth to emphasize that under normal growth conditions, the expression level of *rpoE* in *E. coli* cells is low. Therefore, it was of great interest to compare the activity of the potential *clpB_Li_* promoter (*clpB_Li_P1*) under basal and increased levels of σ^E^. To our knowledge, there has been only one report to date, describing the construction and utility of the *gfp* reporter plasmid for assessing promoter activity in *L. interrogans* [22]. Therefore, due to the still existing limitations of genetic tools that could be easily utilized for examining gene expression in *L. interrogans*, experiments were performed in *E. coli* cells. A modified two-plasmid system has been employed that has been previously successfully used for the identification of many σ^E^-cognate promoters from *E. coli* and also from other bacteria in studies that demonstrated the suitability of this system for assessing promoter activity in *E. coli* [23,24]. Briefly, this system uses two compatible plasmids. One of them is pAC-*rpoE4* carrying the *E. coli rpoE* gene under the control of an arabinose inducible p_BAD_ promoter, and the second plasmid carries a reporter gene fused to a potential σ^E^ promoter. We used the *V. harveyi luxAB* gene, encoding luciferase, as a reporter cloned into a low-copy pGB2 plasmid under control of the tested *clpB_Li_P1* promoter (p*clpB_Li_P1-luxAB*). The method used assumes that shortly after arabinose addition σ^E^ factor overproduced from pAC-*rpoE4* interacts with the *E. coli* RNA polymerase core enzyme forming functional RNA polymerase holoenzyme that is capable of recognizing σ^E^ -controlled promoters located upstream of a reporter gene carried on the second plasmid; in our study, it was the p*clpB_Li_P1-luxAB* plasmid carrying the *luxAB* reporter genes. As shown in Figure 3, the activity of luciferase in both, the exponential and stationary phase of bacterial growth was indeed significantly higher in the presence of the pAC-*rpoE4* plasmid expressing the *E. coli rpoE* (i.e., under σ^E^ excess conditions) than that in cells transformed only with p*clpB_Li_P1*- *luxAB* and producing a basal level of σ^E^.

Importantly, no significant luminescent signal was detected for cells carrying a construct with promoter-less *luxAB* genes. These results strongly suggested that the activity of *clpB_Li_P1* is dependent on σ^E^. Furthermore, we examined *clpB_Li_* expression driven by the *clpB_Li_P1* promoter in *E. coli* cells under heat shock exposure at 42 °C and also under σ^E^ excess conditions (in the presence of pAC-*rpoE4*). Thus, the main aim of this experiment was to compare the *clpB_Li_P1* promoter behavior under thermal stress and σ^E^ overproduction, by monitoring the amount of ClpB_Li_ that was produced. Such an approach was impossible in case of experiments with *pclpB_Li_P1-luxAB* because luciferase is thermolabile and would not withstand the heat shock conditions. To this end, Western blotting analysis of whole-cell lysates of the *E. coli* MC4100Δ*clpB* strain carrying pClpB_L*i*_ alone or together with pAC-*rpoE4* was carried out using anti-ClpB_Li158–334_ serum [15]. Expression of *E. coli rpoE* was induced by the addition of 0.02% arabinose for 30–90 min, while the expression of pClpB_Li_ was induced by exposure to 42 °C for 1 h. As shown in Figure 4A, thermal stress was not sufficient to induce the ClpB_Li_ production (96-kDa protein) in the cells carrying the *clpB_Li_* gene under the control of the *clpB_Li_P1* promoter (see lanes 5 and 6). This is in agreement with our results of primer extension analysis (see Figure 1A) that did not show the presence of the *clpB_Li_* transcripts in *E. coli* cells after heat shock stress. Consistent with our previous results [25], the use of pGB2-ClpB_Li_ [25], which carries the heat-inducible promoter σ^32^ from *E. coli* located upstream of *clpB_Li_*, resulted in increased synthesis of ClpB_Li_ after heat treatment (see lanes 3 and 4). The most interesting observation pertains to the *clpB_Li_* expression under σ^E^ excess conditions (Figure 4B), as the appearance of the 96-kDa ClpB_Li_ in the *E. coli* cells 30–90 min after arabinose addition (see lanes 3–5) provides further evidence that *clpB_Li_P1* is σ^E^ dependent. As can be seen in Figure 4B, there was one more protein that occurred at a higher amount after σ^E^ induction. It is likely that this protein corresponds to a shorter form of ClpB_Li_ (80 kDa) that either represents transcription initiation from the internal promoter or exclusively translation initiation from the internal in-frame ATG codon.

Together, the results obtained with *E. coli* σ^E^ support that ECF σ^E^ factor may mediate the *clpB_Li_* transcription in *L. interrogans* cells.

### 2.3. Role of Leptospiral ECFσ Factors in ClpB_Li_ Gene Expression

It has been demonstrated that σ^E^ contributes to heat shock and oxidative stress response in a number of bacteria and is also involved in the regulation of virulence genes and virulence-associated genes in many bacterial pathogens as well [26]. Since *L. interrogans rpoE* expression was reported to be up-regulated in *Leptospira* cells exposed to elevated temperatures, it is possible that leptospiral σ^E^ factors also regulate expression of genes required for virulence or pathogenesis [13]. This correlates well with a specific function of ClpB in *L. interrogans* and ClpB’s importance during infections [15,17,25]. Of the ten genes encoding ECF σ (σ^E^) factors listed in Table 2, *LIC_12757* (*LA0876*), and *LIC_10559* (*LA3652*) were found to be 1.5- and 2-fold up-regulated at higher temperatures, respectively [13]. Therefore, we considered their products as potential candidates ECF (σ^E^) that transcribe *clpB_Li_* and we evaluated their effect on the transcriptional activity of *clpB_Li_P1.* To this end, we replaced the *E. coli rpoE* gene in the previously employed two-plasmid system with either *LIC_12757* or *LIC_10559* genes. Additionally, *LIC_10144*, which also encodes ECF σ factor (see Table 2) but is not up-regulated under heat shock, was also cloned and used in our assay. As shown in Figure 5, products of the *LIC_12757* and *LIC_10144* genes elevated the transcriptional activity of *clpB_Li_P1* in both the exponential and stationary phase of bacterial growth, supporting our previous observations obtained for σ^E^ from *E. coli* and indicating that an ECF σ^E^ factor is indeed involved in *clpB_Li_* regulation (see Figure 3). Of note, *clpB_Li_P1* displays a much higher activity in the presence of *LIC_12757*. This result suggests that its product may play a key role in *clpB_Li_* expression, especially under stressful conditions like high temperature. To our surprise, *LIC_10*559, whose expression level in response to thermal stress was reported to be more increased than that of *LIC_12757* [13], did not significantly affect the transcriptional activity of *clpB_Li_P1*.

Interestingly, comparison of promoter transcriptional activity in cells harboring p*clpB_Li_P1-luxAB* alone or together with *LIC_10144* or *LIC_12757* genes, as well as with *E. coli rpoE*, revealed that *clpB_Li_P1* is more active in the presence of excess σ^E^ in the stationary growth phase than in the mid-exponential phase when compared to basal σ^E^ level (see Figure 3 and Figure 5). Similar effect has been previously observed and carefully investigated for the *E. coli rpoH* promoter [27,28] and also for other σ^E^-controlled promoters [29]. It has been proposed that an increase in activity of σ^E^ promoters during stationary phase results from elevation in the ppGpp level in this growth phase. To examine whether ppGpp could have an impact on the *clpB_Li_* promoter’s transcriptional activity and, therefore, on *clpB_Li_* expression, the activity of luciferase was determined in both the wild-type strain (MG1655) and its derivative, i.e., the Δ*relAΔspoT* mutant strain lacking ppGpp. We observed (Figure 6) that in the Δ*relAΔspoT* mutant strain, the luciferase activity was significantly lower in the stationary phase than in the wild-type strain, while the opposite was observed in the logarithmic phase of growth. This observation suggests that the *clpB_Li_P1* promoter may be differently regulated by ppGpp during growth—it is repressed in the exponential phase and activated in the stationary phase. It is known that ppGpp is a key factor of the stringent response, which is a widespread response to changing environmental conditions found in all bacterial species tested so far [30]. Importantly, ppGpp is necessary for bacterial virulence and pathogenicity [31]. Presence of a single Rel-like bifunctional protein with (p) ppGpp-hydrolase/synthase activity, Rel_Lin_, in *L. interrogans* points to the existence of the stringent response in this bacterium [32] and corroborates our findings. Still, although ppGpp regulation of *clpB_Li_P1* in *Leptospira* is very likely, it cannot be excluded that the promoter transcriptional activity observed in *E. coli* does not reflect regulation in *Leptospira*. The contribution of the stringent response to *Leptospira* zoonotic lifecycle is yet to be examined.

## 3. Materials and Methods

### 3.1. Bacterial Strains and Plasmids

The bacterial strains used were: *E. coli* MC4100Δ*clpB*:kan supplied by A. Toussaint (Université Libre de Bruxelles, Brussels, Belgium), *E. coli* MG1655, its *relAspoT* derivative (the ppGpp^0^ strain, CF) [33] and *L. interrogans* serovar Copenhageni strain B42 [26]. Plasmids pGB2 [34], pJET1.2 blunt vector (Fermentas, Vilnius, Lithuania), pAC7, pAC-rpoE4, p*σ^3^*^2^-*clpB_Li_* (pGB2-*clpB_Li_*) and pLucVh containing the luxAB (luciferase) genes from *Vibrio harveyi* [21] were used. Plasmids pAC7 and its derivative pAC-*rpoE4* were obtained from J. Kormanec (Institute of Molecular Biology, Slovak Academy of Sciences, Bratislava, Slovak Republic). Both plasmids bear a chloramphenicol resistance gene, and pAC-*rpoE4* also carries the *E. coli rpoE* gene under the control of a p*BAD* promoter [23]. Plasmid p*σ^32^*-*clpB_Li_* is a pGB2 derivative containing the *L. interrogans clpB* gene under the control of the *E. coli* σ^32^-dependent promoter and bearing a spectinomycin resistance gene [25].

### 3.2. Cloning and PCR Methods

To generate a transcriptional fusion between the *clpB_Li_* promoter region (500 bp upstream of the first ATG codon) and luciferase reporter genes, the *V. harveyi luxAB* genes were amplified from pLucVh [21] by PCR using Pfu Turbo DNA polymerase (Agilent Technologies/Perlan Technologies, Warsaw, Poland) with appropriate primers (see Table 3). First, the PCR product was cloned into a pJET1.2 blunt vector, then digested with appropriate restriction enzymes and ligated with the linearized pGB2 *Nde*I-*Hind*III vector and the *clpB_Li_* promoter region to produce the *clpB_Li_P1-luxAB* fusion. The sequence of the resulting construct was confirmed by DNA sequencing (Genomed S.A., Warsaw, Poland). To construct a negative control reporter plasmid (pSD_Li_*luxAB;* promoter-less *luxAB*), *V. harveyi luxAB* genes were amplified together with the Shine–Dalgarno sequence of *clpB_Li_* from p*clpB_Li_P1-luxAB* by PCR using Pfu Turbo DNA polymerase (Agilent Technologies) with appropriate primers (see Table 3). Next, the same cloning procedure was used as described above.

To study the *clpB_Li_* expression in *E. coli* MC4100Δ*clpB* cells, *clpB_Li_* (2583 bp) and its promoter region (500 bp) were amplified by PCR from genomic DNA of *L. interrogans* (extracted with a QIAamp DNA mini kit (Qiagen, Germantown, MD, USA), by using Pfu Turbo DNA polymerase (Agilent Technologies) with appropriate primers (see Table 3). Subsequently, the same cloning procedure was used as described above for the construction of transcriptional fusion.

To clone leptospiral genes encoding ECF σ factors, *LIC_10144*, *LIC_10559*, and *LIC_12757*, into pAC7 under control of a pBAD promoter, the appropriate fragments of genomic DNA isolated from *L. interrogans* were amplified by PCR using Pfu Turbo polymerase and appropriate primers (see Table 3). Then, the same cloning procedure was used as described above for the construction of transcriptional fusion.

### 3.3. RNA Isolation and Primer Extension Assays

Total RNA was isolated from *L. interrogans* and *E. coli* MC4100Δ*clpB* cells carrying the *clpB_Li_* gene along with a region 500 bp upstream of the first ATG codon, using the Total RNA Mini Plus kit (A&A Biotechnology). *L. interrogans* strain was grown in liquid Ellinghausen McCollough Jonhson and Harris medium (EMJH), as described previously [17]. *E. coli* strain was cultured in the LB medium supplemented with 50 µg/mL spectinomycin and 30 µg/mL kanamycin at 30 °C to OD_600 nm_ of 0.35, then transferred to 42 °C for 30 min (heat shock), whereas the thermally non-induced bacteria were further grown at 30 °C. The primers used for primer extension analyses are described in Table 3. Briefly, 2 μg of total RNA were combined with 0.6 pmol of a primer labeled at the 5’-terminus with [γ-^32^P]ATP (specific activity 6000 Ci/mmol; Hartmann) using polynucleotide kinase (Promega), denatured at 75 °C for 10 min, and annealed at 58 °C for 20 min. This was followed by the addition of extension buffer (1x AMV reverse transcriptase buffer (Promega, Medison, WI, USA), 1 mM dNTPs, 20 U RNase-In (Promega), and 10 U of the AMV reverse transcriptase (Promega). The final reaction volume was 25 μL. After incubation at 42 °C for 30 min, 2.5 μg of RNase H (Promega) were added, and incubation was continued at 37 °C for 15 min. cDNA was precipitated in ethanol with 0.35 M sodium acetate (pH 5.5). The samples were then resuspended in a loading buffer (95% formamide, 0.05% bromophenol blue, 0.05% xylene cyanol) and resolved on a 7 M urea, 8% polyacrylamide sequencing gel, run in parallel with a sequencing reaction performed with the same labeled primer and pClpB_Li_ plasmid DNA (see Section 2.1), using a cycle sequencing kit (Jena Bioscience, Jena, Germany).

### 3.4. Luciferase Activity Assay

*V. harveyi* luciferase activity assay was performed as described previously [21], namely, 200 µL culture aliquots were withdrawn at the indicated times and mixed with 7 µL of 10% n-decanal (Sigma/Merck KGaA, Darmstadt, Germany) in ethanol for up to 1 min. Luminescence produced by the enzyme was monitored using a Berhold luminometer.

### 3.5. Detection of ClpB_Li_ in E. coli MC4100ΔClpB Mutant Strain

The pClpB_Li_ (p*P1clpB_Li_*) plasmid, alone or together with pAC-*rpoE4*, was introduced into *E. coli* MC4100Δ*clpB* mutant cells, and *clpB_Li_* expression was explored. Cells carrying p*σ^32^*-*clpB_Li_* (pGB2-ClpB_Li_) were used as a control for the heat-inducible expression. To detect ClpB_Li_ in *E. coli* cultures, Western blotting was performed as described by [35] using anti-ClpB_Li158-334_ serum [15], a peroxidase-coupled goat anti-rabbit secondary antibody (Sigma), and visualized with a chromogenic substrate, 3,3’-diaminobenzidine tetrachloride (DAB, Sigma) and 30% H_2_O_2_.

## 4. Conclusions

To our knowledge, this is the first study providing insight into the transcriptional regulation of *clpB* in the pathogen *L. interrogans*. Primer extension analysis, in combination with promoter transcriptional activity assays permitted us to identify a major heat-inducible promoter located upstream of the *clpB* encoding sequence. Our results clearly indicate that the *clpB_Li_* transcription is under control of one of the σ^E^-type factors, which have been also reported to regulate the biosynthesis of virulence factors in some bacterial pathogens. It has to be noted that ClpB_Li_ is among the known leptospiral virulence factors. It is likely that the product of *LIC_12757* plays a key role in *clpB* transcription in *L. interrogans* cells. Furthermore, we show that the ppGpp alarmone, which also allows expression of virulence genes in some bacterial pathogens, may mediate *clpB* expression in *Leptospira* cells. We believe that findings presented in this study help to improve our knowledge of regulatory mechanisms used by *Leptospira* and may also have an impact on the understanding of the leptospiral virulence mechanisms that allow this pathogen to adapt to its host organisms.

## Figures and Tables

**Figure 1 ijms-20-06325-f001:**
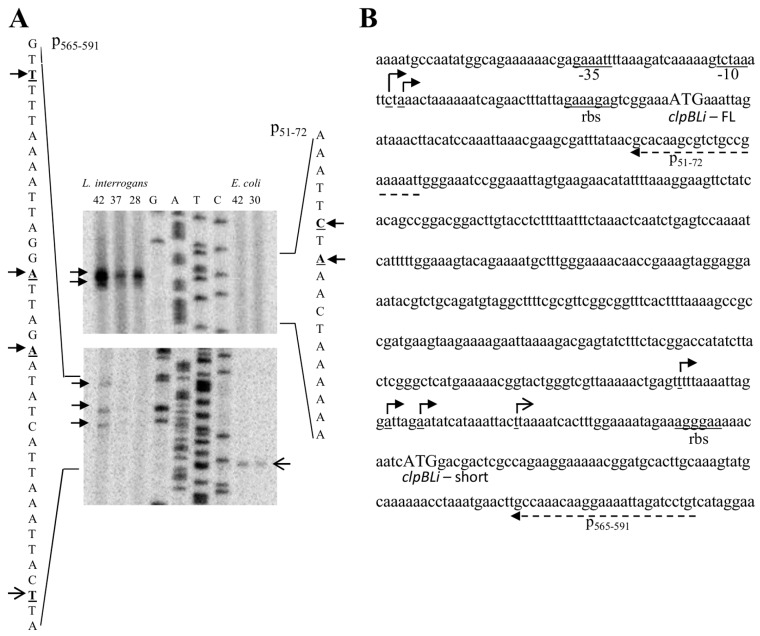
Determination of *clpB_Li_* transcriptional start sites by primer extension assays. (**A**) Primer extension results for the p_565–591_ (left) and p_51–72_ (right) primers, and the corresponding sequencing ladders. Arrows indicate mapped transcriptional start sites; closed arrowheads—*L. interrogans*, opened arrowheads—*E. coli*. The temperature at which bacteria were grown prior to RNA isolation is indicated (28, 30, 37, or 42 °C). (**B**) *clpB_Li_* promoter and coding region with the *clpB_Li_P1* promoter region underlined; transcriptional start sites are indicated by arrows as in (A), primer annealing sites and putative *rbs* sequences are also annotated; *clpBLi*-FL: start codon yielding ClpB95; *clpBLi*-short: internal ATG site producing ClpB80.

**Figure 2 ijms-20-06325-f002:**
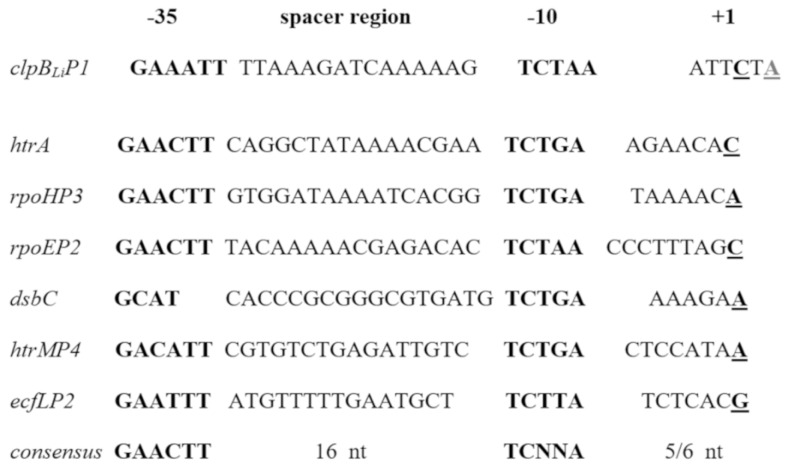
Nucleotide sequence alignment of the well-characterized *E. coli* σ^E^-dependent promoters and the potential *clpB_Li_* promoter (*clpB_Li_P1*). The two transcriptional start sites are marked in bold and underlined, and the major transcriptional start site is denoted by +1. The consensus sequence for the σ^E^-regulated promoter element is shown below the alignment.

**Figure 3 ijms-20-06325-f003:**
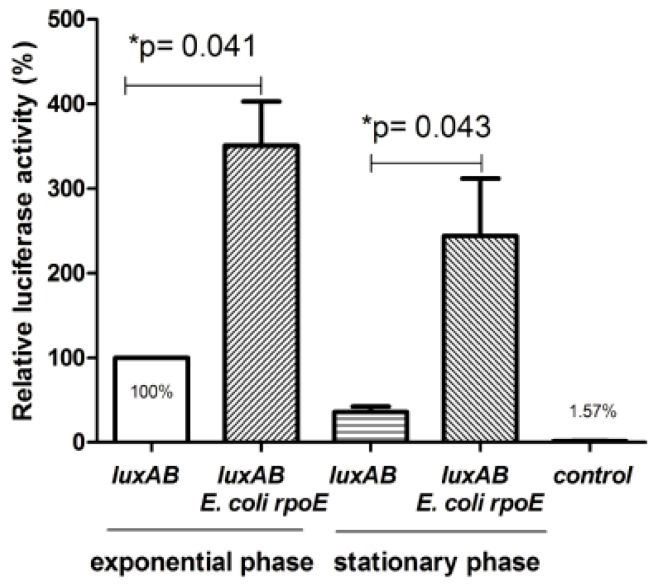
Activity of the *clpB_Li_P1* promoter in *E. coli* cells during the exponential and stationary phase of growth. *E. coli* MG1655 cultures carrying the *clpB_Li_P1-luxAB* (*luxAB*) transcriptional fusion, or a two-plasmid system (*pclpB_Li_P1- luxAB* and pAC-*rpoE4*—*E. coli rpoE*) were grow in LB at 30 °C to OD_600 nm_ of 0.45–0.5 and expression of *rpoE* was induced by addition of 0.02% arabinose for 1 h (for mid-exponential-phase culture) or 6 h (for early-stationary-phase culture), and the luciferase reporter assay was carried out. *E. coli* MG1655 cells carrying pSD_Li_*luxAB* (promoter-less *luxAB*) together with pAC*-rpoE4* were used as a negative control. The luciferase activity (relative luminescence units/OD_600 nm_) in the *E. coli* strain without pAC-*rpoE4* is taken as 100%; bars represent the percentage of luciferase activity normalized to that in the *E. coli* strain without *rpoE* overexpression; the results are presented as the average of three independent experiments, each performed with duplicate cultures, with standard deviations indicated. The paired *t*-test result: * *p* < 0.05 calculated with GraphPad Prism software.

**Figure 4 ijms-20-06325-f004:**
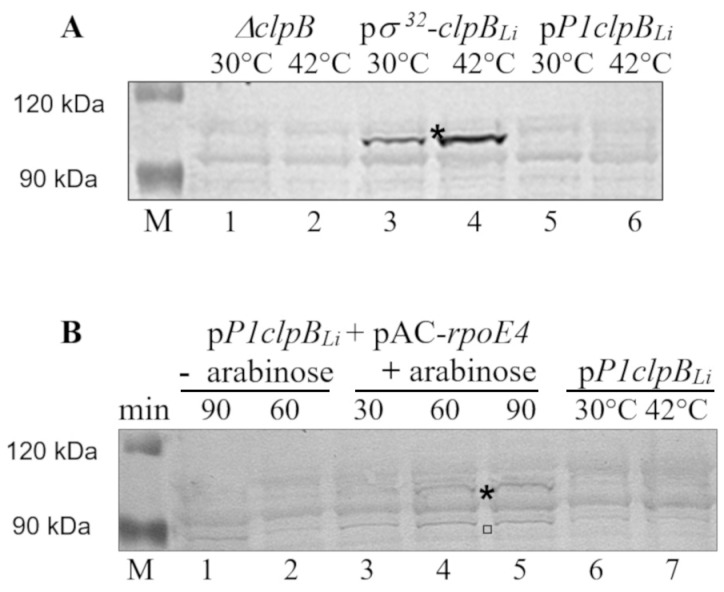
*ClpB_Li_* expression in *E. coli* MC4100Δ*clpB*[pClpB_Li_] cells both under heat shock and σ^E^ excess conditions. (**A**) Immunodetection of ClpB_Li_ with specific antibodies in bacterial lysates of the *E. coli ΔclpB* cells carrying pClpB_Li_ (marked as p*P1clpB_Li_*) and grown at 30 °C, and after 1 h of heat shock at 42 °C. Cells carrying p*σ^32^*-*clpB_Li_* (pGB2-ClpB_Li_; [25]) were used as the control for the heat-inducible expression, while the Δ*clpB* mutant cells without a plasmid were the negative control. (**B**) Immunodetection of ClpB_Li_ in the lysates of *E. coli ΔclpB* [pClpB_Li,_ pAC-*rpoE4*] cells grown at 30 °C both, in the absence and presence of 0.02% arabinose. An asterisk indicates ClpB_Li_ (96 kDa), while a short form of ClpB_Li_ (80 kDa) is marked by a square. The positions of protein size markers (M), the PageRuler prestained protein ladder (Thermo Scientific), are also shown.

**Figure 5 ijms-20-06325-f005:**
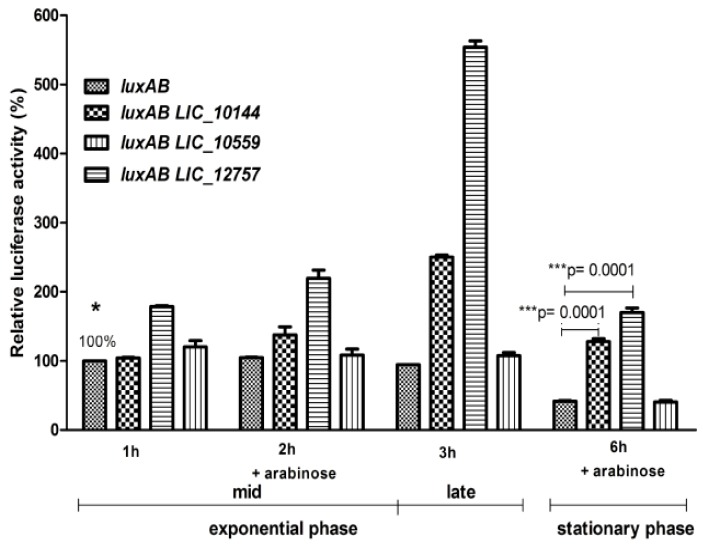
Transcriptional activity of the *clpB_Li_P1* promoter in the presence of leptospiral ECF σ (σ^E^) factors during the exponential and stationary phase of growth. *E. coli* MG1655 cultures carrying the *clpB_Li_P1-luxAB* (*luxAB*) transcriptional fusion or a two-plasmid system (*pclpB_Li_P1-luxAB* and an appropriate *LIC* gene coding σ^E^ factor) were grow in LB at 30 °C to OD_600 nm_ of 0.45–0.5 and expression of *LIC* genes was induced by addition of 0.02% arabinose for 1–3 h (for exponential-phase culture) or 6 h (for early-stationary-phase culture), then the luciferase reporter assay was carried out. The luciferase activity (relative luminescence units/OD_600 nm_) in the *E. coli* strain without *LIC* genes and 1 h after arabinose addition is taken as 100% (*); bars represent the percentage of luciferase activity normalized to that in the *E. coli* strain without *σ^E^* overexpression (*); the results are presented as the average of three independent experiments, each performed with duplicate cultures, with standard deviations indicated. The paired *t*-test result: *** *p* < 0.001 calculated with GraphPad Prism software.

**Figure 6 ijms-20-06325-f006:**
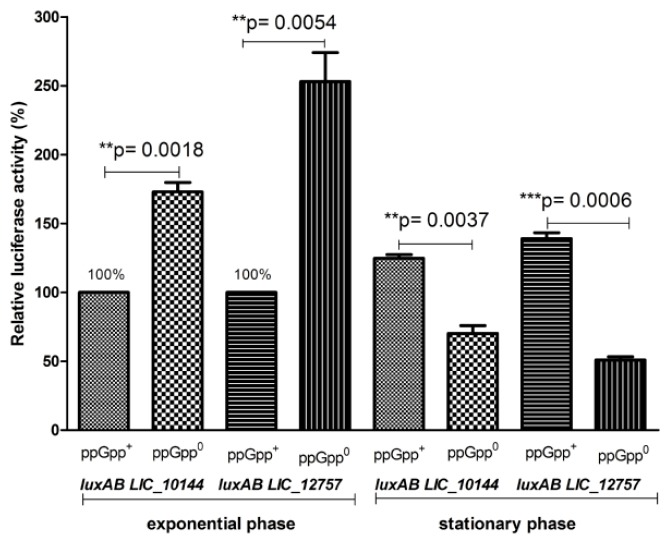
Transcriptional activity of the *clpB_Li_* gene promoter is dependent on ppGpp during the entry of bacteria into the stationary phase. *E. coli* wild-type (MG1655; ppGpp^+^) and Δ*relAΔspoT* mutant (ppGpp^0^) cultures carrying p*clpB_Li_P1-luxAB* along with *LIC_10144* or *LIC_12757* genes were grow in LB at 30 °C to OD_600 nm_ of 0.45–0.5 and expression of the *LIC* genes encoding ECF σ^E^ was induced by addition of 0.02% arabinose for 1 h (for mid-exponential-phase culture) or 6 h (for early-stationary-phase culture), and then the luciferase reporter assay was carried out. The luciferase activity (relative luminescence units/OD600nm) in the *E. coli* ppGpp^+^ cells during exponential phase is taken as 100%; bars represent the percentage of luciferase activity normalized to that of *E. coli* ppGpp^+^ cells during exponential phase; results are presented as the average of three independent experiments, each performed with duplicate cultures, with standard deviations indicated. The paired *t*-test result: ** *p* < 0.01 and *** *p* < 0.001 calculated with GraphPad Prism software.

**Table 1 ijms-20-06325-t001:** Sigma factors identified in *E. coli* and predicted in *L. interrogans* (based on [10,11]).

Species.	σ Factor	Function/Controlled Genes
*L. interrogans*	σ^70^	housekeeping genes (>1000 genes)
	σ^28^	genes encoding components of the endoflagellum and the flagellin-specific chaperone FliS
	σ^24^/σ^Ε^ (ECF σ factor)	extracytoplasmic function (469 putative binding sites in *the Leptospira* genome), stress response and virulence (*clpB*, this study)
	σ^54^	genes encoding putative lipoproteins and the ammonium transporter AmtB
*E. coli*	σ^70^	housekeeping genes
	σ^S^/σ^38^	stationary phase gene expression
	σ^32^/σ^Η^	heat shock response
	σ^28^/σ^F^	flagellar genes/motility genes
	σ^24^/σ^Ε^ (ECF σ factor)	extracytoplasmic function, cell surface stress response, resistance to heat shock and other environmental stresses
	σ^19^/σ^FecI^ (ECF σ factor)	extracytoplasmic function
	σ^54^/σ^Ν^	nitrogen metabolism genes

**Table 2 ijms-20-06325-t002:** ECF σ factors (σ^E^) from *Leptospira interrogans*.

Gene ID ^a^	Protein Accession Number	Number of Amino Acids	Identity/Similarity ^b^ (%)
*LIC_10144*	AAS68777	174	26.0/50.3
*LIC_10225*	AAS68853	301	20.8/51.6
*LIC_10386*	AAS69009	182	20.1/55.6
***LIC_10559***	AAS69180	181	28.3/58.7
*LIC_10644*	AAS69265	174	26.8/61
*LIC_11817*	AAS70405	184	26.1/55.7.9
*LIC_12490*	AAS71055	206	32.7/70.8
***LIC_12757***	AAS71314	180	24.0/55.0
*LIC_13266*	AAS71810	192	31.1/63.3
*LIC_13285*	AAS71829	169	27.6/62.9

^a^ Gene ID was based on ORFs of the genome sequence of *L. interrogans* serovar Copenhageni deposited in GenBank under accession numbers AE016823 (chromosome I) and AE016824 (chromosome II) [12]. ^b^ Identity/similarity scores were determined from sequence alignment of the ECF σ factors from *L. interrogans* serovar Copenhageni and *E. coli* σ^E^ (in GenBank under accession number CDJ72918) using Clustal software. Genes in bold were found to be up-regulated at elevated temperatures [13].

**Table 3 ijms-20-06325-t003:** Oligonucleotides used in this study.

Oligonucleotide	Sequence (5’ to 3’)	Purpose
p_51–72_	ATTTTTCGGCAGACGCTTGTGC	primer extension
p_290–321_	AAGATACTCGTCTTTTAATTCTTTTCTTACTT	primer extension
p_565–591_	ACAGGATCTAATTTTCCTTGTTTGGC	primer extension
*luxANde*I	CATATGAAATTTGGAAACTTCCTTCTC	cloning of luciferase reporter genes
*luxBHind*III	CGACCAAAGCTTACAGTGGTATTTGACGATG	cloning of luciferase reporter genes
*SD_Li_LuxAXma*I	CCCGGGAACTTTATTAGAAAGAGTC	cloning of luciferase reporter genes
*clpB_Li_Nde*I	CATATGAAATTAGATAAACTTACATCCAAATT	cloning of *clpB_Li_*
*clpB_Li_Hind*III	AAGCTTTTAAACTACAACAACTACCTTTCCCT	cloning of *clpB_Li_*
prLi*Xma*I	CCCGGGATAAAATTTCCGAGTCCGATT	cloning of *clpB_Li_* promoter region
prLIC_10144*Nde*I	CATATGGTTCAATCTGATTCTGC	cloning of LIC_10144
prLIC_10144*Hind*III	AAGCTTAGAATTGAAATCCTTGTAG	cloning of LIC_10144
prLIC_10559*Nde*I	CATATGATGCTGAATCCGAATTGC	cloning of LIC_10559
prLIC_10559*Hind*III	AAGCTTTCATTCTTCATAAAATTTCTCC	cloning of LIC_10559
prLIC_12757*Nde*I	CATATGAGCCAAAATTCCGAAAC	cloning of LIC_12757
prLIC_12757*Hind*III	AAGCTTCTATATACTCTCAAAGTCG	cloning of LIC_12757

DNA primers were synthesized by Genomed S.A. (Warsaw, Poland) or Sigma-Aldrich.

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
