# Peer review of "Identification of σE-Dependent Promoter Upstream of clpB from the Pathogenic Spirochaete Leptospira interrogans by Applying an E. coli Two-Plasmid System"

_ijms, 2019, doi:10.3390/ijms20246325_

Round 1
Reviewer 1 Report
This study clearly highlights part of the genetic regulatory mechanisms for virulence of leptospira, allowing a wide panel of receptive species but even why sometimes lab cultures used for challenges failed to reproduce the disease in lab animals.
Author Response
We would like to thank the reviewer for the positive review.
Reviewer 2 Report
The authors have successfully characterized the promoter region of the L. interrogans clpB gene. ClpB is a critical ATP-dependent chaperone belonging to the HSP100 family of AAA+ ATPases, and is highly important for cellular survival under stress conditions. The authors have demonstrates that transcription of clpB begins at two sites approximately 40 bp downstream from the ATG initiation codon. Moreover, transcription was observed to be heat-inducible, which is consistent with the role of ClpB as a heat-shock response element. This work is of interest to the ClpB/Hsp104 community since it examines patterns observed for ClpB homologs from a pathogen aside from the usual E. coli or yeast model systems. Knowledge of how transcription of clpB is regulated by sigma factors is important to understanding cellular regulation of ClpB activity under stress conditions.
Concerns
Throughout the manuscript, the authors use the terms “activity” and “promoter activity.” Though some instances are easily deciphered from context clues, it is not always clear what activity the authors are referring too. This is highly important because a quick reading may lead some readers to conclude that the authors are actually looking at ClpB chaperone activity. This manuscript only examines clpB transcription, but does not examine the function of the gene product. If the authors wish to continue using this language, I recommend that the application of the term “activity” be made more specific to clarify what is meant. The authors mention that two different isoforms of ClpB are observed (pg. 5). Can you comment on what is known for the function of these two isoforms that are 96- and 80-kDa? Is chaperone activity different? Or is this potentially a way to regulate ClpB chaperone function via protein-protein interactions? The authors need to clarify certain experimental details for the reader. For example, experiments were performed with se in either limiting or excess conditions (pg. 5). Please clarify what you hoped to achieve by doing this. The reader can infer the goals of this experimental design, but it would be helpful to explicitly describe the rationale. On the same page, the authors introduce their modified two-plasmid experimental design. However, little description of the method or how it works is provided. Why were these plasmids chosen? Please explain why and how this method was utilized. In western blotting experiments, antibodies against ClpB were utilized. However, no controls are shown to address potential changes in expression of coli proteins that may also impact findings. Though this E. coli strain lacks clpB, it still harbors other heat-shock proteins such as GroEL, ClpAP, ClpXP, FtsH, etc. The authors should repeat experiments with antibodies against some of these E. coli proteins as controls. Moreover, controls against non-heat inducible proteins will be necessary also.
Author Response
Response to Reviewer #2
General comments: The authors have successfully characterized the promoter region of the L. interrogans clpB gene. ClpB is a critical ATP-dependent chaperone belonging to the HSP100 family of AAA+ ATPases, and is highly important for cellular survival under stress conditions. The authors have demonstrates that transcription of clpB begins at two sites approximately 40 bp downstream from the ATG initiation codon. Moreover, transcription was observed to be heat-inducible, which is consistent with the role of ClpB as a heat-shock response element. This work is of interest to the ClpB/Hsp104 community since it examines patterns observed for ClpB homologs from a pathogen aside from the usual E. coli or yeast model systems. Knowledge of how transcription of clpB is regulated by sigma factors is important to understanding cellular regulation of ClpB activity under stress conditions
Concerns
1. Throughout the manuscript, the authors use the terms “activity” and “promoter activity.” Though some instances are easily deciphered from context clues, it is not always clear what activity the authors are referring too. This is highly important because a quick reading may lead some readers to conclude that the authors are actually looking at ClpB chaperone activity. This manuscript only examines clpB transcription, but does not examine the function of the gene product. If the authors wish to continue using this language, I recommend that the application of the term “activity” be made more specific to clarify what is meant.
Response: We used the term “activity” in various combinations such as: promoter activity, activity of luciferase, activity of the clpBLiP1 promoter, activity of clpBLiP1 or disagregase activity (in the case of the clpB gene product, the ClpB molecular chaperone). To avoid any doubts, we added in the revised manuscript the term “promoter transcriptional activity”. We hope that in each case the term “activity” was clarified.
2. The authors mention that two different isoforms of ClpB are observed (pg. 5). Can you comment on what is known for the function of these two isoforms that are 96- and 80-kDa? Is chaperone activity different? Or is this potentially a way to regulate ClpB chaperone function via protein-protein interactions?
Response: As suggested by the Reviewer, in the revised manuscript, we added additional information about the function of the two ClpB isoforms; the revised text reads (lines: 139-145): It has been demonstrated that the two isoforms of ClpB from E. coli cooperate in reactivation of aggregated proteins to form a highly efficient chaperone system [21]. It is very likely that the functional cooperation between ClpB isoforms arises from interactions between them, because it was found that ClpB95 and ClpB80 associate into hetero-oligomers, which boost the aggregate-reactivation potential of the ClpB chaperone [21]. Thus, it was postulated that E. coli produces two isoforms of ClpB to optimize its disagregase activity.
3. The authors need to clarify certain experimental details for the reader. For example, experiments were performed with se in either limiting or excess conditions (pg. 5). Please clarify what you hoped to achieve by doing this. The reader can infer the goals of this experimental design, but it would be helpful to explicitly describe the rationale.
Response: As suggested by the Reviewer, in the revised manuscript, we added some experimental details, such as: (1) lines 179-183: To investigate whether the sE factor may be really involved in the clpBLi transcriptional regulation, promoter activity assay was carried out under sE-limiting (basal) and sE excess (overproduction) conditions. It is worth to emphasize that under normal growth conditions, the expression level of rpoE in E. coli cells is low. Therefore, it was of great interest to compare activity of the potential clpBLi promoter (clpBLiP1) under basal and increased levels of sE.
4. On the same page, the authors introduce their modified two-plasmid experimental design. However, little description of the method or how it works is provided. Why were these plasmids chosen? Please explain why and how this method was utilized. In western blotting experiments, antibodies against ClpB were utilized.
Response: As suggested by the Reviewer, we tried to add some additional information about an E. coli two-plasmid system and the used method; the text now reads (lines: 190-198): Briefly, this system uses two compatible plasmids. One of them is pAC-rpoE4 carrying the E. coli rpoE gene under the control of an arabinose inducible pBAD promoter, and the second plasmid carries a reporter gene fused to a potential sE promoter. We used the V. harveyi luxAB gene, encoding luciferase, as a reporter cloned into a low-copy pGB2 plasmid under control of the tested clpBLiP1 promoter (pclpBLiP1-luxAB). The method used assumes that shortly after arabinose addition sE factor overproduced from pAC-rpoE4 interacts with the E. coli RNA polymerase core enzyme forming functional RNA polymerase holoenzyme that is capable of recognizing sE -controlled promoters located upstream of a reporter gene carried on the second plasmid, in our study it was the pclpBLiP1-luxAB plasmid carrying the luxAB reporter genes. And also lines 218-227: Furthermore, we examined clpBLi expression driven by the clpBLiP1 promoter in E. coli cells under heat shock exposure at 42 °C and also under sE excess conditions (in the presence of pAC-rpoE4). Thus, the main aim of this experiment was to compare the clpBLiP1 promoter behavior under thermal stress and sE overproduction, by monitoring the amount of ClpBLi that was produced. Such an approach was impossible in case of experiments with pclpBLiP1-luxAB because luciferase is thermolabile and would not withstand the heat shock conditions. To this end, Western blotting analysis of whole-cell lysates of the E. coli MC4100DclpB strain carrying pClpBLi alone or together with pAC-rpoE4 was carried out using anti-ClpBLi158-334 serum [15]. Expression of E. coli rpoE was induced by addition of 0.02% arabinose for 30-90 min, while expression of pClpBLi was induced by exposure to 42º C for 1 h.
5. However, no controls are shown to address potential changes in expression of coli proteins that may also impact findings. Though this coli strain lacks clpB, it still harbors other heat-shock proteins such as GroEL, ClpAP, ClpXP, FtsH, etc. The authors should repeat experiments with antibodies against some of these E. coli proteins as controls. Moreover, controls against non-heat inducible proteins will be necessary also.
Response: In this case, we could not agree with the Reviewer’s suggestion. In our opinion, the presence of other heat-shock proteins, including GroEL, ClpXP (ClpA is not induced by heat shock) or FtsH, does not affect clpB expression in E. coli cells, its product level, i.e. the molecular chaperone ClpB. As shown in Fig. 1, for RNA extracted from the E. coli cells no extension product was detectable with the p51-72 primer, indicating that E. coli either does not possess an appropriate factor for clpBLi expression or this factor is not induced in E. coli by a temperature stress. Thus, determination of heat shock proteins by Western-blotting is unnecessary. Further, sE does not affect those heat shock genes in E. coli cells. Heat shock genes in E. coli are mainly dependent on the alternative sigma factor RpoH (σ32). We hope we included the sufficient controls in our study: (1) E. coli DclpB mutant was used for analysis of the clpB gene expression; (2) cells carrying ps32-clpBLi (pGB2-ClpBLi) were used as the control for the heat-inducible expression, while the DclpB mutant cells without a plasmid were the negative control; (3) E. coli DclpB cells carrying pClpBLi and pAC-rpoE4 grown at 30 °C in the absence of arabinose for 60 and 90 min were also used as controls.
Round 2
Reviewer 2 Report
The manuscript is now acceptable for publication.